# Validation of the Measuring Protocol for the Infraspinatus Muscle with M-Mode Ultrasound in Asymptomatic Subjects. Intra- and Inter-examiner Reliability Study

**DOI:** 10.3390/diagnostics13040582

**Published:** 2023-02-04

**Authors:** Marina Ortega-Santamaría, María-Eugenia Torralbo-Álvarez-Novoa, Juan-Nicolás Cuenca-Zaldívar, Fermin Naranjo-Cinto, Samuel Fernández-Carnero, Daniel Pecos-Martín

**Affiliations:** 1Grupo de Investigación en Fisioterapia y Dolor, Facultad de Medicina y Ciencias de la Salud, Departamento de Enfermería y Fisioterapia, Universidad de Alcalá, 28801 Alcalá de Henares, Spain; 2Primary Health Center “El Abajón”, Las Rozas de Madrid, 28231 Madrid, Spain; 3Research Group in Nursing and Health Care, Puerta de Hierro Health Research Institute—Segovia de Arana (IDIPHISA), 28222 Madrid, Spain

**Keywords:** ultrasound, M-mode, infraspinatus, reliability, physical therapy

## Abstract

M-mode ultrasound is a reliable and valid instrument for assessing muscle activity. However, it has not been studied in any of the muscles belonging to the shoulder joint complex, particularly in the infraspinatus muscle. The aim of this study is the validation of the infraspinatus muscle activity measurement protocol with the M-mode ultrasound in asymptomatic subjects. Sixty asymptomatic volunteers were evaluated by two physiotherapists who were blinded; each of them carried out three measurements with the M-mode ultrasound in infraspinatus muscle and analysed the muscle thickness at rest and contraction, velocity of muscle activation and relaxation and Maximum Voluntary Isometric Contraction (MVIC). Intra-observer reliability was significant in both observers, showing good thickness at rest (ICC = 0.833–0.889), thickness in contraction (ICC = 0.861–0.933) and MVIC (ICC = 0.875–0.813); moderate in the activation velocity (ICC = 0.499–0.547) and relaxation velocity (ICC = 0.457–0.606). The inter-observer reliability also had good thickness at rest (ICC = 0.797), thickness in contraction (ICC = 0.89) and MVIC (ICC = 0.84); poor in relaxation time variable (ICC = 0.474) and not significant at the activation velocity (ICC = 0). The muscle activity measurement protocol of the infraspinatus muscle measured with M-mode ultrasound has been found to be reliable in asymptomatic subjects, for both the intra-examiner and inter-examiner.

## 1. Introduction 

Muscle activity evaluation is usually determined by muscle electrical activity measurements using invasive or superficial electromyography. Nevertheless, in the past few years, the use of musculoskeletal ultrasound imaging has been introduced into the process of muscle function assessment; also called Rehabilitative Ultrasound Imaging (RUSI) [1,2]. The use of an ultrasound image (UI) has been shown to provide similar quantifiable information to electromyographic tests in determining muscle activity [3,4,5] in different muscle groups [3,4,6,7,8]. In addition to being accessible and non-invasive, UI enables image capturing to obtain real-time samples of muscular structures, allowing for the correct visualisation of the structure under study (B-mode) [9,10], and the simultaneous measurement of the muscular function during a demanding activity by assessing the displacement of the connective tissue within muscles over time at a high temporal resolution (M-mode) [7,11]. Moreover, M-mode can be used as biofeedback for motor control work [11], as a guiding tool in clinical decision making, and for an improved understanding of muscle and connective tissue adaptations during movement [5,11].

In both research and clinical practice, the use of this method needs validation and a reliability analysis [10] because the UI is operator-dependent and the measurement protocol carried out can affect the results [12]. Ultrasound M-mode has been validated in several muscle groups [3,4,5,6,7], but its use has not been demonstrated in all muscle groups, such as those belonging to the shoulder joint complex. Infraspinatus is one of the rotator cuff muscles that provides stability to the shoulder joint [13]. In particular, this muscle produces external rotation (ER) of the glenohumeral joint [14]. In the frontal plane, this muscle produces compression and slips the humeral head down [15]. In the horizontal plane, it coactivates along with the subscapularis, controlling the humeral head axial rotation and anteroposterior translation movements [16]. In the presence of infraspinatus weakness, a greater activation of the posterior deltoid would be suggested due to its fibre’s disposal. Consequently, an anterior translation of the humeral head appears during ER movement [16], which we will consider controlling during the protocol performance [17].

Secondly, it has been seen that in the measurement of muscle activity, muscle thickness increases along with the activation level during an isometric contraction [18].

When establishing a protocol, we should know whether voluntary muscle recruitment is sequential [19], not necessarily homogenous [20,21] and may differ between fascicles within the same muscle [22], such as the infraspinatus [23,24]. Therefore, for a direct comparison, the UI signals must be obtained from the same sample and muscular region.

We currently know of no studies on shoulder musculature that were carried out where the M-mode ultrasound was used to determine problems in the functionality of these structures. Shoulder pain is the third most common cause of medical consultation [25], after lower-back and cervical pain; since muscular-related problems are the most common causes of this pain, a validation of a measurement protocol using the M-mode ultrasound in the infraspinatus muscle has been proposed during its maximum voluntary isometric contraction (MVIC). Since this muscle has an important contribution to the shoulder joint complex-biomechanics, this protocol determines the most appropriate area to measure the activity of the infraspinatus, making this protocol a valid and reliable tool that has a place and applicability in the clinical field of physiotherapy.

The main objective of this study is to validate a measurement protocol of the infraspinatus muscle activity using M-mode ultrasound through an intra- and inter-examiner reliability study in asymptomatic subjects.

## 2. Materials and Methods

An observational, cross-sectional intra-examiner and inter-examiner reliability study was carried out according to the Standards for Reporting of Diagnostic Accuracy Studies (STARD) [26].

The recruitment took place using informative posters placed in the facilities of the Alcala University, where the measurements were carried out. Subjects interested in participating had to be over 18 years of age, with no shoulder pain for at least the past year, nor affliction of this joint, the neck or upper extremities.

They had to be able to perform a 90° shoulder abduction and at least 30° of glenohumeral ER and preserved muscle strength (grade 5) in both upper limbs (MMSS), in prone position and without shoulder pain [15,27]. It was essential that they had voluntarily signed the informed consent for participation in the study. Exclusion criteria included: past trauma or surgery to the shoulder or neck [16]; signs of cervical radiculopathy; radiculitis or cervical spine shunt; evidence of a full-thickness rotator cuff tear; signs of adhesive capsulitis; having received any injection treatment, acupuncture, dry needling or upper limb-strengthening interventions in the previous 6 months [7]; shoulder pain; neuropathic pain, or any neurological or musculoskeletal problem that may interfere, hinder or prevent the subject from following the study protocol, such as performing the glenohumeral ER movement in the position to be assessed [15,17,28] and systemic diseases including arterial hypertension, heart disease, pacemaker carriers, pulmonary diseases, neurological diseases, mental illness or cognitive impairment, morbid obesity, uncontrolled epilepsy, fever or known pregnancy.

After determining the study population and applying the inclusion and exclusion criteria described above, a randomisation of the sample was carried out to determine the upper limb on which measurements were taken in each study subject, and randomisation of the examiners to determine which one performed the measurement in each participant.

### 2.1. Ethical Considerations

The study was carried out according to the guidelines approved by the Animal Research and Experimentation Ethics Committee (CEI-EA) of the University of Alcalá, and in compliance with the standards set out in the Declaration of Helsinki of the World Medical Association, as updated in 2013 and the Recommendations for the Preparation, Submission, Editing and Publication of Academic Papers in Medical Journals (ICMJE Recommendations), updated in 2018 [29].

### 2.2. Data Collection and Analysis

The measurement protocol of the infraspinatus muscle with M-mode ultrasound.

Position of the patient:

The patient was positioned in a prone position on a hydraulic stretcher, with their head lodged in the facial gap. Both upper limbs maintained 90° of shoulder abduction and the elbow flexed, then dropped to either side of the stretcher. The inflated cuff of the Stabilizer Pressure Biofeedback Unit (Chattanooga Group Inc, Hixson, TN, USA) was placed under the humeral head of the upper limb to control glenohumeral axial rotation [17], and the pressure exerted by the shoulder on the Stabilizer could be varied by +/−2 mm Hg [15,17]. According to a previous study, this significantly and selectively increases the activity of the infraspinatus muscle, its thickness and contraction ratio, while that of the posterior deltoid decreases [15]. The Stabilizer pressure indicator was placed in the subject’s field of vision and remained in the position described above. To maintain the horizontality of the arm to be assessed, a towel was placed under the distal third of the humerus as a fulcrum (Figure 1).

The same instructions were given by both evaluators to all study subjects. There was prior familiarisation of the subjects with the movement to be performed.

Probe position:

The probe was held in a fixed position over the measuring point, established by manual grasping by the investigator. The minimum necessary pressure was exerted for good visualisation of the structures on the ultrasound scanner [30].

To determine the measurement site, the B-mode of the ultrasound machine was used to visualise the following structures: in the transverse axis, the vertebral limit of the scapula spine and the spinoglenoid notch at the axillary limit of the scapula. For this purpose, the anatomical reference had to be visualised on the lateral limits of the ultrasound screen and matched with the limit of the probe. There, a mark was placed on the skin with a dermatographic pencil. Once the midpoint of the line between these references had been measured with a ruler, the probe was changed to the craniocaudal axis. Perpendicular to this reference line, the probe was lowered to locate the scapula inferior angle. Once marked, the midpoint between the latter and the calculated midpoint on the scapula spine was determined, resulting in the Point O measurement for the visualisation of the infraspinatus muscle using the ultrasound M-mode (Figure 2). The Point O was made to coincide with the midpoint of the scapula spine. Point O was matched with the midpoint of the probe, which was placed in a transverse axis, parallel to the infraspinatus fibres.

A Vinno E35. 1.9.51 ultrasound scanner and an X6–16L linear probe were used to evaluate the infraspinatus muscle by ultrasonography. The B-mode of the ultrasound machine was used to determine the optimal point for measuring and imaging the muscle. Once located, the M-mode was activated to record the infraspinatus muscle activity.

The variable representing muscle thickness measured at rest was measured using the M-mode ultrasound, where the distance was taken between the superficial and deep fascial connective tissue of the infraspinatus muscle at rest. Similarly, the variable representing muscle thickness was measured in MVIC towards ER movement. Data were also collected of the activation velocity (the velocity achieved to move from the resting state to the state of maximum contraction) and relaxation velocity (the velocity of the muscle to move from the state of maximum contraction to the final resting state), measured in turn using the M-mode (Figure 3). The MVIC was performed towards the glenohumeral ER with 90° of shoulder abduction in prone position against a digital wireless dynamometer positioned perpendicular to the patient’s forearm. In this way, the maximum isometric force performed by each subject was measured by simple monitoring.

For the validation of the protocol for measuring infraspinatus muscle activity using the M-mode ultrasound, we describe in detail the procedure we carried out to allow its reproducibility.

The ultrasound machine was configured to a pre-set option, where the image was optimised according to the study subject for the best visualisation of the structures. Once the probe was placed at point O, the M-mode was activated to visualise and record infraspinatus muscular activity. Three 5 s MVICs were performed towards ER, leaving 2 min of rest between each contraction to avoid muscle fatigue [31].

The measurement protocol described above was carried out by two assessors. Both were trained and familiarised with the protocol beforehand. They performed the same functions at different points in time. To provide the same measurement conditions, each subject was assessed on the same day, leaving 10 min between one assessor and the other.

The subject was placed back in the measurement position by the next examiner and any marks on the skin that might indicate the measurement site of the previous examiner were removed beforehand, so that the examiners were blinded from the data collected by the other. This same procedure was performed by each examiner between each of the 3 measurements.

### 2.3. Statistical Analysis

For the statistical analysis, the program R Ver. 4.1.3 was used (R Foundation for Statistical Computing, Institute for Statistics and Mathematics, Welthandelsplatz 1, 1020 Vienna, Austria). The qualitative variables were described in absolute values and frequencies and the quantitative variables with mean and standard deviation.

The intra- and inter-observer intraclass correlation coefficient ICC(2, 1) was calculated for the observers’ measurements as relative reliability; defining it as poor (<0.5), moderate (0.5–0.75), good (0.75–0.9) and excellent (>0.9) two-way mixed single measures (consistency/absolute agreement) [32].

The absolute reliability was calculated, which is the standard error of measurement (SEM: SD × √1−ICC). The minimum detectable change (MDC: SEM × √2 × 1.96)) was also calculated.

Finally, the correlation between the average of the ultrasound variables and the clinical–demographic variables was calculated using the Pearson correlation matrix, or by means of a punctual polychoric–polyserial matrix in the case of qualitative variables; defining it as negligible (<0.29), low (0.3–0.49), moderate (0.5–0.69), high (0.70–0.89) and very high (>0.90) [33].

## 3. Results

The study involved 60 healthy volunteer subjects with a similar proportion of men and woman, a BMI of 22.75 ± 2.84, most of whom played sports and had the right arm as their dominant one (Table 1).

The intra-observer reliability was significant (*p* < 0.05) in both observers, being good for the Thickness at rest, Thickness in contraction and MVIC variables; and moderate-low for the Activation velocity and Relaxation velocity variables. The SEM did not exceed 10% with narrow confidence intervals and a small MDC, indicating good measurement precision and an ability to detect small changes (Table 2).

The inter-observer reliability was significant (*p* < 0.05), except for the Activation velocity variable; being good for the Thickness at rest, Thickness in contraction and MVIC variables; and low in the Relaxation velocity variable. The SEM did not exceed 10% with narrow confidence intervals; except for the Relaxation velocity variable with a high MDC, the Thickness at rest and Thickness in contraction variables, which were those with the lowest SEM and higher in accuracy (Table 3).

There were significant positive correlations (*p* < 0.05) in the following comparisons: Activation velocity vs. Age, Activation velocity vs. BMI, Activation velocity vs. Weight, Activation velocity vs. Work activity, MVIC vs. Age, MVIC vs. BMI, MVIC vs. Dominant upper limb, MVIC vs. Height, MVIC vs. Weight, Relaxation velocity vs. Age, Relaxation velocity vs. BMI, Relaxation velocity vs. Weight, Relaxation velocity vs. Work activity, Thickness at rest vs. Age, Thickness at rest vs. BMI, Thickness at rest vs. Height, Thickness at rest vs. Weight, Thickness in contraction vs. Age, Thickness in contraction vs. BMI, Thickness in contraction vs. Dominant upper limb, Thickness in contraction vs. Height, Thickness in contraction vs. Weight and Thickness in contraction vs. Work activity. No significant negative correlations were found for: MVIC vs. Gender, Thickness at rest vs. Gender and Thickness in contraction vs. Gender (significant results are shown in red), with the following correlations:Negligible in MVIC vs. Gender, MVIC vs. Height, Relaxation velocity vs. BMI, Relaxation velocity vs. Weight, Thickness at rest vs. Age, Thickness at rest vs. Gender, Thickness in contraction vs. Gender and Thickness in contraction vs. Height comparisons.Low in Activation velocity vs. Age, Activation velocity vs. BMI, Activation velocity vs. Weight, Activation velocity vs. Work activity, MVIC vs. Age, MVIC vs. BMI, MVIC vs. Weight, Relaxation velocity vs. Age, Relaxation velocity vs. Work activity, Thickness at rest vs. BMI, Thickness at rest vs. Height, Thickness at rest vs. Weight, Thickness in contraction vs. Age, Thickness in contraction vs. BMI, Thickness in contraction vs. Dominant upper limb, Thickness in contraction vs. Weight and Thickness in contraction vs. Work activity.Moderate in the comparison of MVIC vs. Dominant upper limb.There is neither a high nor a very-high correlation (Appendix A).

## 4. Discussion

This is the first study to attempt to quantify infraspinatus muscle function with M-mode ultrasound, designing a measurement protocol so that it can be carried out by other examiners in their clinical practice, since intra-examiner and inter-examiner reliability analysis has been found to be good. Intra-examiner reliability was slightly higher than inter-examiner reliability.

The best results were obtained for the following variables: Thickness at rest, Thickness at contraction and MVIC; where they were found to have good reliability, both for the same examiner and between examiners.

Ultrasound sonography has been studied and validated as a reliable tool for the measurement of infraspinatus muscle thickness, comparing it with magnetic resonance imaging [34]. Likewise, in 2015, Koppenhaver et al. spoke about the importance and benefits of RUSI as a tool in the field of physiotherapy. As in the present study, he measured infraspinatus muscle thickness, obtaining high intra-observer and inter-observer reliability [10]. However, the study used B-mode ultrasound where the movement of connective tissue over time is not as clear as it is in the M-mode, validated for the first time by Dieterich et al. in the gluteus medius and gluteus minimus [7].

On the other hand, referring to the measurement protocol used, Koppenhaver et al. measured the infraspinatus muscle thickness by making a longitudinal cut to the fibres of this muscle from the superomedial border of the scapula spine [10]. This region of the infraspinatus, according to Kuwahara et al. who studied the changes in mechanical properties of the infraspinatus muscle tissue in its different regions, appears to play a greater role in the glenohumeral abduction movement than in ER [23]. Therefore, the researchers of this study wanted to validate a measurement protocol where coincidentally it was measured in the region of the muscle where it seems to participate to a greater degree in the glenohumeral ER [23], the main function of the infraspinatus muscle [14].

Regarding inter-observer reliability, the results were similar to those obtained by Vasseljen et al. in their studies of deep lumbar multifidus and abdominal muscles [4], as well as Romero-Morales et al., in their study of soleus muscle [6]; using the M-mode, both obtained good and excellent inter-observer reliability, respectively. This is why we support the clinical relevance associated with obtaining good inter-observer reliability; and according to Wu Wei Ting et al., UI is a reliable tool—one of the most widely used for the evaluation of shoulder disorders [35], and is not dependent on the evaluator or their experience [36].

Muscle activation is usually measured using electromyography. This provides information about the recruitment of motor units in the maintenance of muscle tone and contraction. However, surface EMG can be interfered with by the activity of other muscles close to the measuring site [6]. Authors such as Vasseljen et al., Scarlata et al. and Dieterich et al. have studied the association between muscle activation and movement in muscles other than the infraspinatus, establishing a clear relationship between the two variables [3,7,8]. However, there is variability in this correlation, where authors such as Tweedle et al. have not obtained the same results, indicating that the UI can detect the onset of muscle activation before the EMG does [37]. Moreover, Vasseljen et al. state that deep muscle activation can be reliably measured with the M-mode if a good quality ultrasound scanner is used [3], as we have used in the present study. Thus, for the activation and relaxation velocity variables studied, intra-examiner reliability was moderate. On the other hand, the inter-observer reliability for the relaxation velocity variable was low and the activation velocity variable was poor and insignificant. We propose that the swipe speed during M-mode scanning should be explored to obtain better results for this muscle. Even so, more research is needed to compare the methods for the measurement of activation and relaxation velocity to other muscles, such as the infraspinatus.

Among the limitations of this study, first is that it was carried out with healthy subjects. It would be interesting to conduct an additional experiment with people experiencing a shoulder pathology or pain, to see whether the results would change or be similar. Secondly, a follow-up and assessment several days later could be considered to increase the reliability of the study. The third limitation is that this protocol was only evaluated in a certain position where a greater and more specific activation of the infraspinatus muscle was observed. However, it would be interesting to evaluate this protocol in other positions where there is also a specific activation of the infraspinatus, and to compare if there are any differences in the variables or in reliability. Another limitation is that the measurement protocol was performed at a specific point of the infraspinatus, not knowing whether the results could be modified at another point within the same muscle since the infraspinatus is a muscle, whose intramuscular architecture is known to be divided into fascicles that vary in shape and contractile capacity [23]. The M-mode ultrasound needs such research to provide a reliable assessment of protocols and procedures.

Future research of interest involves analysing the electromyographic activity of the infraspinatus muscle while simultaneously measuring the thickness change with the M-mode ultrasound. Furthermore, it would be interesting to evaluate the clinical response of the infraspinatus muscle’s thickness changes measured with the M-mode ultrasound in shoulder pathology.

## 5. Conclusions

The protocol for measuring muscle activity of infraspinatus muscle measured with the M-mode ultrasound proved to be reliable in asymptomatic subjects, for both intra-examiner and inter-examiner, with the best results being obtained for the variables of muscle thickness at rest, muscle thickness at contraction, and MVIC. It might be a useful method to assess in pathological contexts in further studies.

## Figures and Tables

**Figure 1 diagnostics-13-00582-f001:**
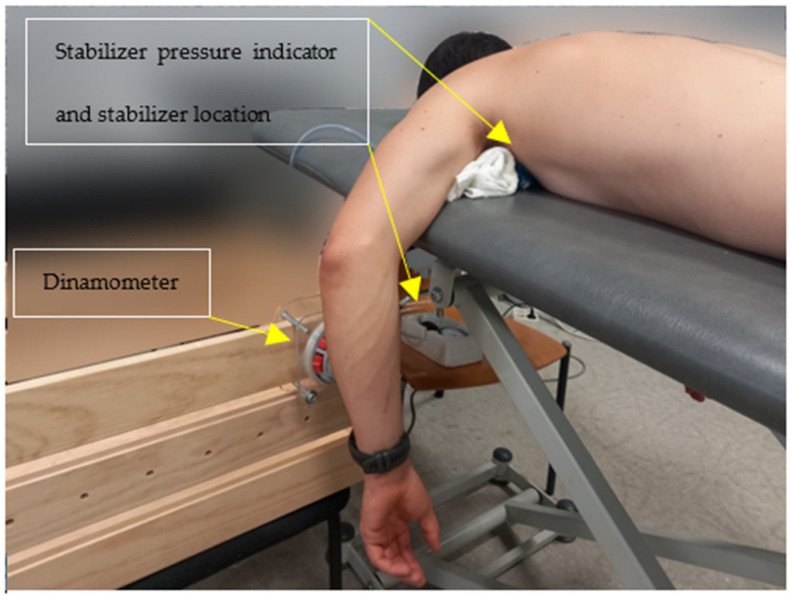
Position of the patient.

**Figure 2 diagnostics-13-00582-f002:**
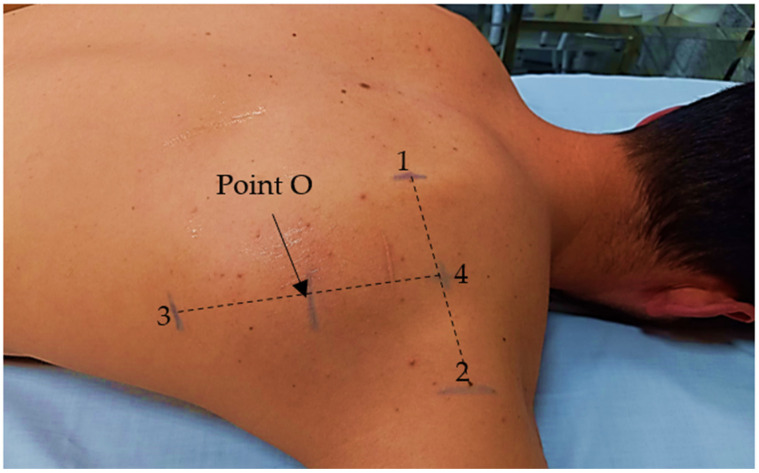
Measurement site protocol: 1: vertebral limit of the scapula spine; 2 spinoglenoid notch; 3: scapula inferior angle: 4: midpoint between 1 and 2; 5: Point O (midpoint between 3 and 4).

**Figure 3 diagnostics-13-00582-f003:**
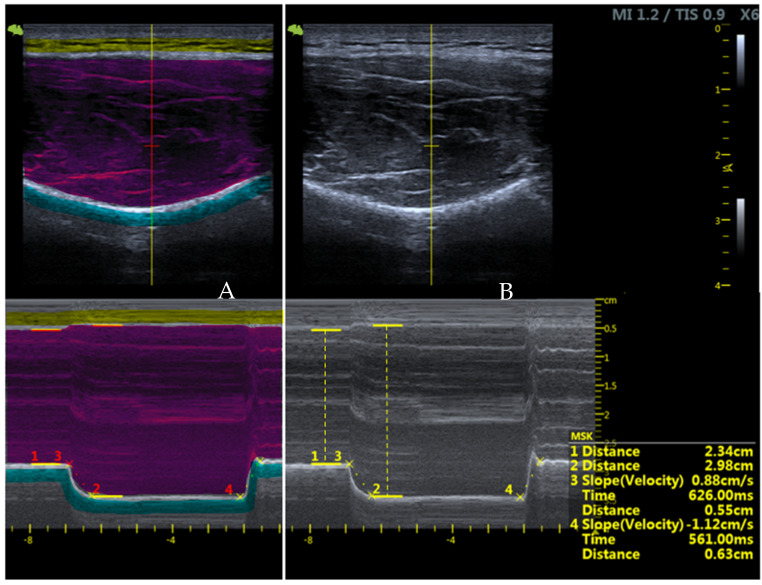
Infraspinatus muscle visualisation using the B-mode at the optimal point for measuring (at the top) and M-mode at the bottom. (**A**) Highlighted the fat tissue (yellow), infraspinatus muscle (red) and bone (blue). (**B**) 1: thickness at rest; 2: thickness at contraction; 3: activation velocity; 4: relaxation velocity.

**Table 1 diagnostics-13-00582-t001:** Clinical and demographic characteristics of the participants.

*n*		60
Gender, *n* (%)	Female	29 (48.3)
	Male	31 (51.7)
Age		26.35 ± 11.60
Weight (kg)		66.93 ± 11.15
Height (m)		1.71 ± 0.09
BMI		22.75 ± 2.84
Neck pain upper limb or shoulder, *n* (%)	No	60 (100.0)
Work activity, *n* (%)	Student	38 (63.3)
	Worker	22 (36.7)
Sport activity, *n* (%)	No	10 (16.7)
	Yes	50 (83.3)
Dominant upper limb, *n* (%)	Left	6 (10.0)
	Right	54 (90.0)
Observed upper limb, *n* (%)	Left	28 (46.7)
	Right	32 (53.3)

Data expressed with mean ± standard deviation or with absolute and relative values (%).

**Table 2 diagnostics-13-00582-t002:** Intra-observer reliability.

	Observer 1				Observer 2			
	ICC (95%CI)	Average Measurement (95% CI)	SEM (95% CI)	MDC	ICC (95% CI)	Average Measurement (95% CI)	SEM (95% CI)	MDC
Thickness at rest	0.833 (0.728, 0.907)	2.053 (1.87, 2.236)	0.182 (0.138, 0.226)	0.505	0.889 (0.847, 0.923)	2.06 (1.932, 2.189)	0.173 (0.149, 0.196)	0.479
Thickness in contraction	0.861 (0.772, 0.923)	2.399 (2.205, 2.593)	0.176 (0.131, 0.221)	0.488	0.933 (0.906, 0.954)	2.421 (2.287, 2.556)	0.137 (0.116, 0.158)	0.380
MVIC	0.875 (0.794, 0.931)	7.545 (6.103, 8.988)	1.244 (0.883, 1.605)	3.448	0.813 (0.745, 0.867)	6.929 (6.039, 7.818)	1.593 (1.321, 1.864)	4.414
Activation velocity	0.547 (0.352, 0.72)	0.382 (0.241, 0.523)	0.273 (0.187, 0.359)	0.758	0.499 (0.373, 0.619)	0.51 (0.423, 0.597)	0.292 (0.236, 0.348)	0.809
Relaxation velocity	0.457 (0.252, 0.654)	0.514 (0.278, 0.75)	0.519 (0.331, 0.707)	1.438	0.606 (0.494, 0.708)	0.517 (0.442, 0.593)	0.211 (0.182, 0.241)	0.586

ICC: intraclass correlation coefficient; SEM: standard error of measurement; MDC: minimal detectable change; 95% CI: 95% confidence interval.

**Table 3 diagnostics-13-00582-t003:** Inter-observer reliability.

	ICC (95% CI)	Average Measurement (95% CI)	SEM (95% CI)	MDC
Thickness at rest	0.797 (0.633, 0.893)	2.05 (1.873, 2.227)	0.203 (0.139, 0.267)	0.563
Thickness in contraction	0.89 (0.792, 0.943)	2.399 (2.208, 2.59)	0.158 (0.111, 0.204)	0.437
MVIC	0.84 (0.705, 0.917)	7.483 (6.042, 8.925)	1.45 (1.013, 1.888)	4.020
Activation velocity	0 (-0.322, 0.329)	0.486 (0.331, 0.641)	0.329 (0.213, 0.444)	0.911
Relaxation velocity	0.474 (0.168, 0.698)	0.513 (0.347, 0.678)	0.338 (0.222, 0.455)	0.938

ICC: intraclass correlation coefficient; SEM: standard error of measurement; MDC: minimal detectable change; 95% CI: 95% confidence interval.

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
