# Peer review of "Validation of the Measuring Protocol for the Infraspinatus Muscle with M-Mode Ultrasound in Asymptomatic Subjects. Intra- and Inter-examiner Reliability Study"

_diagnostics, 2023, doi:10.3390/diagnostics13040582_

Round 1
Reviewer 1 Report
First of all, I would like to thank the editor of this journal for allowing me to review this fantastic research work. I would also like to thank the researchers in Madrid for their work in doing this valuable and good work. The study fits perfectly in the journal and, moreover, I believe that such work is necessary in Physiotherapy to advance in the process of objectifying clinical and health processes.
The study is of high quality, the measurements are well done, the protocol is adequate and well justified. I think the editor should accept this research work so that the readers of this journal can enjoy reading this study as much as I did.
Finally, I would like to comment on some minor details with the aim of slightly improving this work, which in itself is almost perfect.
1-Abstract: If I were an author of this study, I would be more specific in the conclusions of the abstract as to which analyses have inter-examiner reliability and in which analyses inter-examiner reliability is not found.
2-Introduction: The introduction is well written and argued. Congratulations to the authors.
3-Methods: I would only add a little more information in the statistical analysis part. For example, put the formula of the SEM (SDx√1-ICC) or MDC (SEMx√2x1.96), also the statistical test that the authors used for inter- and intra-examiner evaluation (2-wat mixed effects ANOVA, for example) or also that the reliability data are based on the guidelines established by Shrout & Fleiss (>0.75 indicates excellent reliability, etc.) or the author(s) that the researchers relied on.
4-Results: They are excellently written.
5-Discussion: I would only add a justification as to why the authors believe that some tests such as activation speed or relaxation time have such low or even zero inter-rater reliability.
Congratulations to the authors for this excellent work.
End of review
Author Response
Dear reviewer. In red is written in each part.
First of all, I would like to thank the editor of this journal for allowing me to review this fantastic research work. I would also like to thank the researchers in Madrid for their work in doing this valuable and good work. The study fits perfectly in the journal and, moreover, I believe that such work is necessary in Physiotherapy to advance in the process of objectifying clinical and health processes.
The study is of high quality, the measurements are well done, the protocol is adequate and well justified. I think the editor should accept this research work so that the readers of this journal can enjoy reading this study as much as I did.
Finally, I would like to comment on some minor details with the aim of slightly improving this work, which in itself is almost perfect.
1-Abstract: If I were an author of this study, I would be more specific in the conclusions of the abstract as to which analyses have inter-examiner reliability and in which analyses inter-examiner reliability is not found.
Dear reviewer. Many thanks for these words which encourage all the team for future research. The abstract has been reviewed and found this data at lines 22-27 If more information is considered let us know.
2-Introduction: The introduction is well written and argued. Congratulations to the authors.
Thanks for the comment.
3-Methods: I would only add a little more information in the statistical analysis part. For example, put the formula of the SEM (SDx√1-ICC) or MDC (SEMx√2x1.96), also the statistical test that the authors used for inter- and intra-examiner evaluation (2-wat mixed effects ANOVA, for example) or also that the reliability data are based on the guidelines established by Shrout & Fleiss (>0.75 indicates excellent reliability, etc.) or the author(s) that the researchers relied on.
Dear reviewer, thanks for the comment. The SEM and MDC formulas has been included lines 263-264.
The detailed information about the statistical test with the reference has been added see lines 261-262.
4-Results: They are excellently written.
Thanks for the comment.
5-Discussion: I would only add a justification as to why the authors believe that some tests such as activation speed or relaxation time have such low or even zero inter-rater reliability.
Dear reviewer, thanks for the comment. A possibility about this issue has been added at lines 399-400.
Congratulations to the authors for this excellent work.
End of review
Reviewer 2 Report
The paper is well written and it presents enough informations about the proposed methodology. It might be a good starting point to assess the reliability of the tested evaluation method in further studies focused on pathological context, where more variables might be present, influencing the results.
Author Response
The paper is well written and it presents enough informations about the proposed methodology. It might be a good starting point to assess the reliability of the tested evaluation method in further studies focused on pathological context, where more variables might be present, influencing the results.
Dear reviewer. Thanks for the comment. Has been added a note at the end of conclusions lines 431-432